# A Network Landscape of HPVOPC Reveals Methylation Alterations as Significant Drivers of Gene Expression via an Immune-Mediated GPCR Signal

**DOI:** 10.3390/cancers15174379

**Published:** 2023-09-01

**Authors:** Jesse R. Qualliotine, Takuya Nakagawa, Sara Brin Rosenthal, Sayed Sadat, Carmen Ballesteros-Merino, Guorong Xu, Adam Mark, Art Nasamran, J. Silvio Gutkind, Kathleen M. Fisch, Theresa Guo, Bernard A. Fox, Zubair Khan, Alfredo A. Molinolo, Joseph A. Califano

**Affiliations:** 1Department of Otolaryngology—Head and Neck Surgery, University of California San Diego, La Jolla, CA 92093, USA; 2Gleiberman Head and Neck Cancer Center, Moores Cancer Center, University of California San Diego, La Jolla, CA 92093, USA; 3Department of Otorhinolaryngology, Head and Neck Surgery, Graduate School of Medicine, Chiba University, Chiba 263-8522, Japan; 4Center for Computational Biology and Bioinformatics, Department of Medicine, University of California San Diego, La Jolla, CA 92093, USA; 5Earle A. Chiles Research Institute, Providence Cancer Center, Portland, OR 97213, USA; 6Department of Otolaryngology—Head and Neck Surgery, Johns Hopkins University, Baltimore, MD 21218, USA

**Keywords:** network analysis, head and neck squamous cell carcinoma, human papillomavirus, HPV, oropharyngeal neoplasms, epigenetics, exome sequencing, The Cancer Genome Atlas (TCGA)

## Abstract

**Simple Summary:**

In this manuscript, we demonstrated a particular chemokine axis (CXCR3/CXCL9,10,11) to be highly expressed in HPV-associated oropharynx carcinoma (HPVOPC). This classically paracrine chemokine axis is thought to play an anti-tumor role through immune surveillance and tumor suppression. More recently, however, the same axis has also been shown to have paradoxically a pro-tumor effect, whereby in an autocrine signaling fashion it increases tumor cell proliferation, angiogenesis, and metastasis in colorectal, breast, cervical, and gastric cancers, but this is the first time in HPVOPC. An athymic murine model demonstrates that CXCR3 antagonism does indeed inhibit tumor growth; however, CXCR3 antagonism in immunocompetent animals shows mixed results. The CXCR3 axis is implicated as a possible driver of malignancy in HPVOPC via tumor growth stimulation, though caution is warranted as inhibition of this axis may also adversely impact immunosurveillance. Further work is indicated to investigate opportunities for targeted therapy.

**Abstract:**

HPV-associated oropharynx carcinoma (HPVOPC) tumors have a relatively low mutational burden. Elucidating the relative contributions of other tumor alterations, such as DNA methylation alterations, alternative splicing events (ASE), and copy number variation (CNV), could provide a deeper understanding of carcinogenesis drivers in this disease. We applied network propagation analysis to multiple classes of tumor alterations in a discovery cohort of 46 primary HPVOPC tumors and 25 cancer-unaffected controls and validated our findings with TCGA data. We identified significant overlap between differential gene expression networks and all alteration classes, and this association was highest for methylation and lowest for CNV. Significant overlap was seen for gene clusters of G protein-coupled receptor (GPCR) pathways. HPV16–human protein interaction analysis identified an enriched cluster defined by an immune-mediated GPCR signal, including CXCR3 cytokines CXCL9, CXCL10, and CXCL11. CXCR3 was found to be expressed in primary HPVOPC, and scRNA-seq analysis demonstrated CXCR3 ligands to be highly expressed in M2 macrophages. In vivo models demonstrated decreased tumor growth with antagonism of the CXCR3 receptor in immunodeficient but not immunocompetent mice, suggesting that the CXCR3 axis can drive tumor proliferation in an autocrine fashion, but the effect is tempered by an intact immune system. In conclusion, methylation, ASE, and SNV alterations are highly associated with network gene expression changes in HPVOPC, suggesting that ASE and methylation alterations have an important role in driving the oncogenic phenotype. Network analysis identifies GPCR networks, specifically the CXCR3 chemokine axis, as modulators of tumor–immune interactions that may have proliferative effects on primary tumors as well as a role for immunosurveillance; however, CXCR3 inhibition should be used with caution, as these agents may both inhibit and stimulate tumor growth considering the competing effects of this cytokine axis. Further investigation is needed to explore opportunities for targeted therapy in this setting.

## 1. Introduction

Human-papillomavirus-related oropharynx cancer (HPVOPC) is a lethal disease, increasing at epidemic rates [1,2,3]. Viral oncoproteins E6 and E7 have putative roles in transformation through p53 and pRB suppression, respectively. Still, compared to traditional tobacco- and alcohol-associated head and neck squamous cell carcinoma (HNSCC), HPVOPC tumors have relatively few alteration events [4,5]. Traditionally, attention has been largely directed toward genetic alterations, i.e., single-nucleotide variation (SNV), as primary drivers of gene expression. Using tumor whole-exome sequencing, PIK3CA was identified to be the most frequently mutated gene in HNSCC overall, with a subset of HPV-associated tumors having PIK3CA or PIK3R1 as the only mutated cancer gene [6,7]. Genomic mutation analysis identifies the PI3K-AKT-mTOR pathway as frequently dysregulated in HPVOPC [6]; however, networks are dysregulated through other mechanisms, such as splice variant alterations (ASE), genomic copy number variation (CNV), or methylation dysregulation. The goal of this work was to better understand the relative contributions of diverse alteration events toward driving the overall phenotype in HPVOPC.

Network analysis helps integrate tumor alteration events, such as mutations or ASE, within protein–protein gene interaction networks in order to infer cellular pathways and functions affected by tumor alteration, allowing for heterogeneity at the individual gene level [8]. We hypothesize that ASE, methylation, and CNV alterations, in addition to SNV alterations, may be closely correlated with differentially expressed gene (DEG)-based protein network alterations, implicating these alterations as primary drivers of altered gene expression. We used protein–protein network analysis to link alterations in the spliceosome (ASE), methylome (methylation dysregulation), CNV, SNV, and overall gene expression in the context of protein network space, in a discovery cohort of primary tumor samples of HPVOPC, defining key altered pathways. These findings were then validated in a similar tumor cohort from TCGA and allowed identification of novel networks altered in HPVOPC amenable to therapeutic targeting.

## 2. Materials and Methods

### 2.1. Patient Samples for Discovery Cohort

Primary tumor tissue samples were obtained from a test cohort of 46 patients with HPV-related oropharyngeal squamous cell carcinoma (HPVOPC), as described previously [9,10]. In brief, HPV tumor status was determined using HPV in situ hybridization for high-risk HPV subtypes or p16 immunohistochemistry, and all tissues were micro-dissected to yield at least 80% tumor purity. For comparison, normal oropharynx mucosal tissue from uvulopalatopharyngoplasty (UPPP) surgical specimens were obtained from 25 cancer-unaffected controls. Clinical data describing this cohort have been published previously and are available in the Supplementary Table 2 of that publication [9]. All tissue samples were collected from the Johns Hopkins Tissue Core under an approved Institutional Review Board protocol (#NA_00036235), and written informed consent was obtained from each patient prior to the collection of samples.

### 2.2. RNA Preparation and RNA-Seq Analysis

Methods for sequencing and data processing of RNA using the RNA sequencing protocol have been previously described [10,11]. Briefly, tumors and normal samples passed minimum quality thresholds after RNA extraction. Sequencing was performed using the HiSeq 2500 platform sequencer (Illumina, San Diego, CA, USA) and the TruSeq Cluster Kit for 2 × 100 bp sequencing. The RNA sequencing data were then normalized using the version 2 protocols as developed by TCGA [5]. The RNA sequences were aligned to the GRCh37/hg19 genome assembly using MapSplice2 version 2.0.1.9. Supervised clustering of the RNA sequencing data was performed using R/Bioconductor version 3.3.2.

### 2.3. Gene Expression Analysis

As previously described [10], gene expression values were quantified using transcript models based on TCGA methods using RSEM version 1.2.9 and upper quartile-normalized, according to TCGA normalization protocol [5,12]. Gene expression between tumors and normal tissue was compared by log-fold change. Based on the distribution of the log-fold change values among all genes analyzed, log-fold change values that were greater than two standard deviations above or below the median were identified as significant. Next, the overall gene expression was compared between normal tissue and all primary tumor samples using a Wilcoxon test and adjusted for the false discovery rate (FDR) using the Benjamini–Hochberg correction [13].

### 2.4. MBD-Seq DNA Methylation Analysis

Genome-wide DNA methylation analysis was carried out using MBD-seq, similar to what was described previously [9,14,15]. Briefly, DNA was sonicated, end-repaired, and ligated to SOLiD P1 and P2 sequencing adaptors lacking 50 phosphate groups, using the NEBNext DNA Library Prep Set for SOLiD kit (NEB), according to the manufacturer’s recommended protocol. Libraries were then nick-translated with Platinum Taq polymerase and divided into two fractions: an enriched methylated fraction that was subjected to isolation and elution of CpG-methylated library fragments by using MBD2–MBD-bound magnetic beads, as described previously [15], and a total input fraction that was left unenriched. These fractions were then amplified using 4–6 cycles for the total input, and 10–12 cycles for the enriched methylated fractions according to the NEBNext DNA Library Prep Set for SOLiD kit (NEB). The resulting libraries were subjected to emulsion PCR, bead enrichment, and sequencing on a SOLiD sequencer to generate, on average, approximately 25–50 million 50 bp single-end reads per sample, according to the manufacturer’s protocols (Life Technologies, Carlsbad, CA, USA). The resulting color-space reads were aligned using Bioscope software, version 1.2 (Applied Biosystems, Waltham, MA, USA). MACS1 peak identification software (version 1.4.2) was used to identify regions of methylation by comparing the enriched versus total input libraries for each sample, with a default cutoff of *p* < 10^−5^. The methylation status of each 100 bp segment across the genome for each sample in the cohort described above was determined as an indicator of the presence of any intersection of the segment with regions of DNA methylation, as identified by MACS peak calling [16]. For each gene, methylation was analyzed for 1500 bp upstream and 500 bp downstream from the canonical transcriptional start site [17]. The differential methylation was tested in each 100 bp segment within these regions of interest by comparing the numbers of samples with and without a methylation signal in the segment between tumors and normal samples, using Fisher’s exact test, since this test is not susceptible to imbalances between cases and controls. All the methods are available within the “differential.coverage” R package, version 0.2.0. [18,19].

### 2.5. TCGA RNA-Seq and DNA Methylation Data for Validation (Validation Cohort)

Publicly available normalized beta values generated using Illumina Infinium HumanMethylation450 (HM450K) BeadChip, normalized gene expression data obtained by RNA sequencing, and clinical data were downloaded from the Broad TCGA GDAC (http://gdac.broadinstitute.org (accessed on 1 November 2018)).

### 2.6. SNV Analysis: Exome Sequencing, Filtering, and Alignment

As previously described [20], DNA was extracted with the DNeasy Blood and Tissue Kit (Qiagen, Hilden, Germany) for high-quality extraction according to the manufacturer’s instructions. DNA samples from tumors and matched lymphocyte controls were quantified with a Qubit Fluorometer (Thermo Fisher Scientific, Waltham, MA, USA). More than 1 μg of each sample was prepared with a sonication-based library construction and enrichment method according to the Beijing Genomics Institute, as previously described [21]. The prepared DNA libraries were hybridized to an Agilent SureSelect Human All Exon Kit to capture the target exome, and sequencing was executed with the Illumina HiSeq 4000 sequencing system (Beijing Genomics Institute, Shenzhen, China) at a variable depth of 50 to 150×. The exome sequencing pipeline was performed on 92 samples, which included the 46 tumor samples and the 46 normal matched lymphocyte samples. To generate sequence alignment and variant calls, we implemented our exome analysis pipeline on cfncluster (version 1.3.1) [22]. Short reads were mapped to the human 1000 Genomes Project (version 37) [23] with BWA-mem (version 0.7.12). Subsequent processing was performed with SAMtools (version 1.1), Picard Tools (version 1.96), and the Genome Analysis Toolkit (version 2.4-9) [24], and it consisted of the following steps: sorting and splitting of the BAM files, marking of duplicate reads, local realignment, indel realignment, and recalibration of base quality scores. Somatic variants were named with MuTect. Oncotator was used to annotate variants, which were then filtered to include exonic insertions and deletions and non-synonymous variants, with an Exome Aggregation Consortium (ExAC) and 1000 Genomes population allele frequency of less than 0.05.

### 2.7. Splice Variant Identification and Outlier Statistics

For identification of splice variants unique to tumor samples, an algorithm utilizing outlier analysis, as described previously [9], was applied to putative junctions identified from MapSplice alignment output utilizing R version 3.2.1. Junction expression was compiled across all samples, using a value of 0 for samples that did not have the putative junction. Values were then normalized as RPM (reads per million) and log-transformed. Junctions were removed if there was no difference in expression between any tumor and any normal tissue, as well as if junctions were located on X or Y chromosomes. Junctions were then mapped to known genes and exons based on hg19 genome assembly, and selected as putative splice variants if they were identified either as a skip (junction that skips a known exon), insertion (start or end outside a known exon), or deletion (start or end within a known exon), using package org.Hs.eg.db, version 3.4.0 [25]. The junction expressions for selected junctions were normalized by dividing by the total gene expression (using RSEM [12]), and gene alignments were determined by genome assembly or manual curation if overlapping genes were present. Outlier analysis was performed to identify the number of tumors with outlier expression in comparison with the distribution of expression in normal tissue, using functions from the OGSA Bioconductor package [26,27]. Next, the number of outliers occurring in tumors was compared with that in normal tissue using Fisher’s exact test, since this test is not susceptible to imbalances between cases and controls. The *p*-values were FDR-adjusted with the Benjamini–Hochberg method, and adjusted *p*-values below 0.05 were used to determine junctions with significantly higher outliers in tumors compared with normal samples.

### 2.8. Integrative Genome Viewer Confirmation

Putative splice junctions identified as significant through outlier statistics were then visualized using the Integrative Genome Viewer (IGV, Broad Institute, version 2.3) [28], as described previously [9]. From RNA sequencing data, BAM files were loaded into IGV to directly visualize compiled RNA read data. Read data were visualized at the start and end of each junction and compared between normal and tumor samples. Putative junctions were confirmed if the overall gene expression was observed in both normal and tumor tissue, and a unique wild-type splicing event was identifiable in normal tissue. Junctions were then categorized as either an alternative start site, canonical skipping, insertion, deletion, intron retention, alternative end site, intronic short segment (junction confined within one intron), or noncoding (junction start and end site occur within the noncoding region).

### 2.9. CNV Analysis

DNA alignments were processed using CNVkit commands in the order of target, anti-target, coverage, reference, fix, segment, segmetrics, call, and export. Respective parameters were selected as recommended for tumor-only inputs.

### 2.10. Protein Interaction Network

Protein interaction network data were acquired from the STRING database [29]. Only high-confidence edges were used (edges with confidence > 700).

### 2.11. Network Localization

We measured the network localization of genes significantly altered by each omics dataset (SNV, splicing, methylation, CNV, expression) by counting the number of edges shared between the genes in each focal set. This is similar to the ‘significance’ measure used in the STRING database [29]. We measured the number of edges connecting a subgraph composed of 80% of focal genes and compared this to a distribution composed of degree-matched random node sets (1000 permutations). To measure an empirical *p*-value, we calculated the localization of the full gene set, and compared this to the distribution of localization on 1000 randomly selected, degree-matched gene sets.

### 2.12. Network Propagation

For each alteration type, comparing HPVOPC to normal tissue, differences were analyzed with network propagation techniques [8], using the method of random walk with restart. Network propagation simulates how heat would diffuse, with loss, through the network by traversing the edges, starting from an initially hot set of ‘seed’ nodes, as follows:F^t^ = αW′F^t−1^ + (1 − α)Y,(1)
where F^t^ is the heat vector at time t, Y is the initial value of the heat vector, and W′ is the normalized adjacency matrix. α is the fraction of total heat retained at every timestep, and is set to 0.5 in our analysis. The resulting heat vector was compared to a null model composed of 5000 randomly selected degree-matched seeds, to create a gene-level proximity z-score. Network proximal genes were defined as genes with proximity z-scores > 2, to identify those genes significantly proximal to the seed genes.

The network propagation algorithm was seeded with genes significantly altered in each data type (splicing, SNV, CNV, methylation, and expression). For hypergeometric tests, values of overlap were reported as −log(*p*). Significant overlap between a pair of networks was denoted by a score > 3 = −log (0.05) for *p* = 0.05.

### 2.13. Validation of Discovery Cohort Results by Comparison with TCGA Cohort

To validate our findings with the discovery cohort of the primary tumor, a similar analysis was performed with data from TCGA using HPVOPC tumors compared to non-malignant head and neck epithelial tissue samples (TCGA Provisional version, updated in 2016, http://cancergenome.nih.gov/ (accessed on 1 November 2018)). These samples’ data included 94 HPV-positive HNSCC and 16 normal tissues. We also downloaded the RNA-Seq by Expectation Maximization (RSEM)-normalized gene expression values for the same samples from the Broad GDAC Firebrowse website (http://firebrowse.org/ (accessed on 1 November 2018)). To identify the specific splice variants, we conducted the alignment of TCGA RNA sequencing data using MapSplice [30] to the GRCh37/hg19 genome assembly. Splice junction data from the alignment were extracted for the following analysis. We normalized junction expression values as RPM (reads per million) and performed log transformation. The junctions that had no difference in expression between tumor and normal samples were filtered out. The junctions that mapped to X, Y, and mitochondrial chromosomes were also filtered out. All the junctions which mapped to known genes and exons based on the GRCh37/hg19 genome assembly were considered as putative splicing events. These ASE were identified either as a skip (junction that skips a known exon), insertion (junction that starts or ends outside a known exon), or deletion (junction that starts or ends within a known exon). Expression values of these selected junctions were normalized by the RSEM values for the genes that were downloaded from TCGA. We performed an outlier analysis to identify the significant junctions between tumor and normal samples.

### 2.14. HPV–Human Protein–Protein Interaction

We obtained the HPV–human protein–protein interaction map, which included 2988 interactions between 12 HPV proteins and 2061 human proteins, as published by Farooq et al. [31]. We analyzed only the 2434 HPV16 protein interactions since greater than 90% of HPV-related HNSCC is associated with HPV16. The STRING database (https://string-db.org (accessed on 1 February 2021)) was used to search for protein interactions and Cytoscape (https://cytoscape.org (accessed on 1 February 2021)) was used to visualize the interactions.

### 2.15. Immunohistochemistry (IHC) for CXCR3 and CXCL9 Expression in Primary HPV-Associated Oropharynx Tumors

Eight p16-positive oropharyngeal cancer and eight normal tonsils were used for IHC. Precise IHC methods were shown previously [32]. IHC sections, 5 μm-thick, were melted for two hours at 55 °C. All sections were deparaffinized with xylene and gradually hydrated with 100%, 95%, and 70% ethanol. After rinsing the ethanol with abundant distilled water, antigen activation was performed for 45 min in a steamer (>90 °C) using IHC Antigen Retrieval Solution (Invitrogen, Carlsbad, CA, USA). Slides were cooled, washed with distilled water, and endogenous peroxidase was blocked using BLOXALL blocking solution (Vector Laboratories, Burlingame, CA, USA). Sections were then incubated overnight with primary antibody for CXCR3 (#26756-1-AP, 1:100, Thermo Fisher Scientific) and CXCL9 (#701117, 1:200, Thermo Fisher Scientific), and for 30 min with anti-rabbit biotinylated secondary antibody (#BA-1400, 1:400, Vector Laboratories, Burlingame, CA, USA) or anti-mouse biotinylated secondary antibody (#BA-1400, 1:400, Vector Laboratories, Burlingame, CA, USA), at room temperature. The ABC Kit (Vector Laboratories, Burlingame, CA, USA) was used as the detection system, and 3,3′-diaminobenzidine (Vector Laboratories, Burlingame, CA, USA) as the chromogenic source. Mayer’s hematoxylin was used for counterstaining. After washing with distilled water, sections were gradually dehydrated with 70%, 95%, and 100% ethanol and xylene and mounted in mounting medium.

### 2.16. Multiplex Immunohistochemistry (mIHC)

Eight p16-positive oropharyngeal cancer and eight normal tonsils were used for mIHC. Formalin-fixed, paraffin-embedded block tissue microarrays with 4 μm-thick tissue sections were used. Tissue slides were deparaffinized, subjected to heat-induced epitope retrieval, and stained with the Leica Bond RX (Lecia Biosystems, Singapore) Autostainer. A multiplex IHC panel was performed on the tissue slides using the following antibodies: anti-CD68, anti-CD163, anti-CD16, and anti-cytokeratin (Table 1). Tissue slides were incubated with DAPI as a counterstain and cover-slipped with VectaShield mounting medium (Vector Labs). Control tissue samples were stained separately for each marker. Hematoxylin and eosin staining was performed on each sample and reviewed by a pathologist to ensure the representativity of the tissue sample. mIHC tissue slides were scanned on a PhenoImager (Akoya Biosciences, Marlborough, MA, USA) in qptiff format. Images were captured at a 0.5 µm pixel resolution using the 20× objective with saturation protection as a whole-slide overview.

### 2.17. Single-Cell RNA Sequencing

We extracted count matrices of single-cell RNA-seq (scRNA-seq) of HPVOPC data from previous work by Cillo et al. [33]. We assigned cell types based on the k = 18 clustering solution in the Single-Cell Expression Atlas (https://www.ebi.ac.uk/gxa/sc/experiments/E-GEOD-139324/results/tsne?colourBy=metadata&metadata=organism_part&geneId=ENSG00000138755 (accessed on 1 February 2021)). Analyses based on t-SNE plots and feature plots were also performed using the Single-Cell Expression Atlas, and 18 clusters were annotated with comprehensive immune cell markers: T cell; CD3D, CD4Tcell; CD4, CD8Tcell; CD8B, B cell; CD79B, regulatory T cell; FOXP3, NK cell; CD56 (NCAM1), macrophage; CD68, monocyte cell; CD14. Scatter plots of the expression levels of CXCL9/CXCL10/CXCL11 for each cluster using each single-cell datapoint were visualized using the ggplot2 package in R version 4.0.3.

### 2.18. Mouse Model and Reagents

In this work, we employed two mouse models: an immunodeficient model using athymic nude mice implanted with 1.5 × 10^6^ UM-SCC-47 cells (HPV16-positive head and neck squamous cell carcinoma cell line), and an immunocompetent model using C57BL/6 mice and 1.5 × 10^6^ mEER cells (squamous cell carcinoma cell line isolated from the oropharyngeal epithelium of a C57BL/6 mouse, with cells graciously obtained from Dr. Chad Spanos’s Lab at the University of South Dakota/Sanford Research Center, Vermillion, SD, USA). For each experiment, sixteen female mice (6 weeks) were implanted with the appropriate tumor cells in 100 μL of Corning Matrigel Matrix (CORNING, Corning, NY, USA), subcutaneously, in the flank. Mice were closely monitored and euthanized when a necrotic tumor was observed and/or when the mice lost 20% or more of their initial weight. Two days before tumor implantation, the mice received intraperitoneal injections of 5 ug/g body weight of the CXCR3 antagonist AMG 487 (Tocris Biosciences, Bristol, UK) in the vehicle: 5% of hydroxypropyl-beta-cyclodextrin solution (sigma-Aldrich, Saint Louis, MO, USA), or received the vehicle daily for the entire span of the experiment (n = 8 for both conditions). After the completion of the experiment at day 25 or 26, the mice were euthanized, and tumors were harvested.

## 3. Results

Gene expression was compared between tumors and normal tissue within our discovery cohort. At a threshold of adjusted *p* < 0.005 and absolute log-fold change greater than 2.0, 738 genes were identified as significantly differentially expressed. Of these 738 genes, 549 were found in the STRING protein interactome. Similarly, for other alteration classes, 23 genes showed differential alteration for SNV, 289 for methylation, 77 for splicing, and 520 for CNV. First, we assessed the overlap of each alteration type among these collections of genes. A hypergeometric test was used to calculate the likelihood of finding the observed number of overlapping nodes by chance. All genes in the interactome were used as the background set for the hypergeometric calculation. This analysis demonstrated that there was highly significant overlap between the genes for DEG-splicing and DEG-methylation (−log(*p*) = 13; *p* = 2 × 10^−6^ for both) and between CNV and SNV genes (−log(*p*) = 3.1; *p* = 0.045) (significant = score > 3 = −log (0.05) for *p* = 0.05) (Figure 1A).

Next, each class of alteration was analyzed using a single round of network propagation. Altered genes within each class of alterations were used as seeds for each network. For DEGs, there were 549 seed genes, and following network analysis, 1396 genes in total were found to be significant within the network space; that is, there were 1396 nodes in network proximity to the seed genes. For SNV, there were 23 seed genes, and 662 genes in total were found to be significantly proximal in the network space. Similarly, for methylation, there were 289 seed genes and 1436 network-proximal nodes, for splicing, 77 seeds and 769 network-proximal nodes, and for CNV, 520 seeds and 1004 network-proximal nodes.

To evaluate the localization of the network gene sets found for each alteration type, the number of edges within gene sets in the network space was compared to that of degree-matched random sets. The *p*-values were calculated by permutation on the un-sampled distribution. Note that this method has more power to detect significance with larger gene sets, because small gene sets have, on average, very few edges connecting their nodes in the interactome. This analysis demonstrated that gene sets in the network space derived from differentially expressed seed genes were statistically different in association from random sets for differential expression, CNVs, and methylation (*p* < 1 × 10^−10^ for all), but not significantly different from SNVs (*p* = 0.19) or splicing alterations (*p* = 0.06) (Appendix A). Of note, the gene set sizes for these two alteration classes were smaller than the sizes for the other alteration classes by an order of magnitude.

To contextualize the interaction among these networks, defined by alteration class, we then looked at the significance of overlapping genes between network classes. Network overlap was then assessed by calculating the overlap between network subgraph pairs (shared nodes). In a similar fashion to the overlap assessment of genes prior to network analysis, a hypergeometric distribution that considers the size of the networks was used to calculate the likelihood of finding the observed number of overlapping nodes by chance. All genes in the interactome were used as the background set for the hypergeometric calculation. This analysis demonstrated significant overlap between networks of DEGs and all alteration classes (Figure 1B). The methylation network had the greatest overlap (−log(*p*) = 96; *p* = 2 × 10^−42^). The SNV and splicing networks were also highly significant (−log(*p*) = 12, *p* = 6 × 10^−6^; and −log(*p*) = 11, *p* = 2 × 10^−5^, respectively), while the CNV network had the least overlap, though still more than would be expected by chance (−log(*p*) = 3.2; *p* = 0.041).

Next, we validated our findings by performing an analogous analysis in HPVOPC and normal tissues in a validation cohort (TCGA). As expected, hypergeometric testing between gene sets from the discovery and validation cohorts demonstrated significantly greater overlap in members of the gene sets than would be expected from random collections for differentially expressed genes and each of the different alteration classes (Figure 1C). Of note, the method for defining the splicing alterations within this validation cohort did not allow for hypergeometric testing. As such, we used a targeted approach for the splicing validation, whereby we analyzed ASE events that were significantly altered in the discovery cohort and in TCGA.

Network analysis was then performed within each dataset, and the overlap of genes within the network space was compared between the discovery and validation cohorts after excluding the seed genes themselves. As shown in Figure 1D, this analysis demonstrated that the overlap in network space genes was much more significant than the pre-network propagation seed genes for methylation, SNV, and splicing alterations. On the other hand, the significance in the overlap between cohorts after network analysis decreased for CNV alterations.

Next, global differential gene expression (DEG) was defined as the end-product of all the tumor alterations to assess the relationship between each of the dysregulated alteration classes and gene product dysregulation, and to investigate how responsible each of these mechanisms are for the overall gene dysregulation in this cancer. To contextualize the specific alterations with this set of DEGs, an analysis was performed to explore the significance threshold of the associations between network space gene sets from DEGs and those from other tumor alteration classes, to help identify any significant network associations. We searched for an enrichment of differentially expressed genes in the network-proximal gene sets of the CNV, methylation, splicing, and SNV seed sets, again using hypergeometric tests to assess significance. We measured the fraction of genes within each network-proximal set that were significantly differentially expressed (absolute magnitude of log-fold change = 1 or greater) for a range of *p*-values. Surprisingly, this analysis demonstrated that splicing alterations, methylation, and SNV all had significant genes in the network space overrepresented in the DEG network at higher significance thresholds (smaller *p*-values), with CNV having the lowest fraction of significant genes in its network space overlapping with that of DEGs, with a log-fold change magnitude of 1 or greater (Figure 1E). The same analysis was also performed with the validation cohort, and similar findings emerged (Figure 1F).

The significance threshold of the association between network space gene sets from DEGs and individual tumor alteration classes were again explored when considering genes whose absolute magnitude of log-fold change was 2 or greater. A similar pattern emerged among tumor alteration classes, in which splicing alterations had the greatest fraction of significant genes in the network space overrepresented in the DEG network at higher significance thresholds, followed by methylation (Appendix A). With larger, but still significant *p*-values, the fractions of significant genes were similar with splicing, methylation, and SNV. Throughout all levels of significance thresholds, CNV had the lowest fraction of significant genes in its network space overlapping with that of differentially expressed genes, with a log-fold change magnitude of 2 or greater.

Network clusters are collections of genes associated with specific types of cellular processes or pathways, as annotated by the KEGG, REACTOME, and Gene Ontology databases. To investigate the specific biologic functions, specific gene clusters were identified in each network-proximal gene set, using a graph-based clustering algorithm [34]. Functional annotation was conducted using the gProfiler tool [35], using the set of network-proximal genes as the background set. We then looked at the network clusters that are driving this association between specific alteration classes and differentially expressed genes. For example, considering the higher overlap between genes found within the splicing alteration and the overall differential gene expression networks for highly significant genes with log-fold change > 2, we were interested in whether clusters of the included genes could describe larger cellular pathways that were contributing to this signal. An enrichment analysis created clusters of nodes representing cellular pathways, and a similar analysis was performed for the network of each tumor alteration type. Finally, the overlap in network clusters was visualized with a heatmap to describe significant pathways and GO terms found to be significant within both the DEG network and the other tumor alteration networks (Figure 2A). Significant overlaps were most frequent between methylation and DEGs. Interestingly, there was a relative lack of association between networks of SNV and DEGs for clusters of genes, except for “immune system”, as demonstrated by the gold rectangles, while other alternative splicing and methylation classes appeared to drive several other clusters of pathways, especially those representing G protein-coupled receptor (GPCR) signaling pathways, “G alpha (i) signaling events” (pink rectangles) and “G alpha (q) signaling events” (blue rectangles). Clusters including “G alpha (i) signaling events” had significant overlap between the DEG network and splicing, as well as methylation alteration networks. Clusters including “G alpha (q) signaling events” had significant overlap between the DEG network and methylation alteration networks. Interestingly, neither cluster including “G alpha (i) signaling events” or “G alpha (q) signaling events” featured significant overlap in the network space between DEG and SNV alterations.

To validate the overlap between DEG network clusters and each alteration network cluster in the discovery cohort, we performed the same analysis using TCGA validation cohort (Figure 2B). “G alpha (i) signaling events”, “G alpha (q) signaling events”, and “immune system” were also overlapped between the DEG cluster and the methylation cluster in the validation cohort. “Immune system” was also overlapped between the DEG cluster and the SNV alteration cluster. Since each cluster contains many pathways and GO terms, to capture the overlap in pathway and GO term levels more accurately and to identify the most significant pathways among the overlapped clusters, we created bubble plots to visualize each annotated pathway and its relative significance among the overlapped clusters. To do this for each pathway (e.g., G-alpha (i) signaling events), the negative log of the *p*-value for the significance of being differentially altered in splicing, methylation, or SNV clusters was plotted against that for the DEG cluster. This was performed for both the discovery cohort (Figure 2C) and the validation cohort (Figure 2D). Among the pathways of G-alpha (i), G-alpha (q), and immune system, G-alpha (i) had a higher association in both the DEG cluster and methylation clusters in the discovery cohort.

Therefore, we focused on GPCR signaling, as two different GPCR signaling clusters were included in overlapping pathways between differentially expressed genes and splicing and methylation data in the discovery and validation cohorts (Figure 2C,D). To elucidate the interaction between HPV oncoproteins and GPCR signaling, we used publicly available data defining HPV–human protein–protein interactions using mass spectrometry and yeast 2 hybrid assays [31]. We examined the data looking at proteins associating with HPV16 proteins, as greater than 90% of HPVOPC are associated with HPV16. In total, HPV16 E1, E2, E4, E5, E6, E7, L1, and L2 were selected, and we visualized the interactions between HPV proteins and human proteins. This resulted in a total of 2434 HPV16–human protein interactions (L1: 24; L2: 105; E1: 44; E2: 437; E4: 21; E5: 427; E6: 675; E7: 701; Figure 3A). Based on these HPV proteins and human proteins and considering our previous results highlighting a strong GPCR signal for both ASE and methylation alterations (Figure 2C,D), we combined 526 known GPCR signaling proteins and re-analyzed the protein–protein interactions of these using the STRING database and visualized them using Cytoscape (Figure 3B,C and Appendix A). In particular, chemokine signaling was highly associated with HPV viral proteins E2, E4, and E5, and viral oncoproteins E6 and E7.

To confirm the expression pattern of chemokine signaling in both the discovery and validation cohorts, we examined genes included in GPCR signaling pathways from differentially expressed genes of both cohorts. The discovery cohort included 173 genes from GPCR signaling pathways and the validation cohort included 105, and 39 genes were included in both cohorts (Figure 3D,E). The chemokine signaling pathway comprised 39 and 27 genes from both the discovery and validation cohorts, respectively, and 14 genes were shared between both cohorts. Among these 14 genes, we found that *CXCL9*, *CXCL10*, and *CXCL11* were differentially upregulated in both cohorts.

CXCL9, CXCL10, and CXCL11 are ligands of CXCR3 and have been increasingly recognized as key regulators of the tumor microenvironment [36,37]. This classically paracrine chemokine axis is thought to play an anti-tumor role through immune surveillance and tumor suppression [38,39]. More recently, however, the same *CXCL9*, *10*, and *11/CXCR3* axis has also been shown to have paradoxically a pro-tumor effect as well, whereby in an autocrine signaling fashion, it increases tumor cell proliferation, angiogenesis, and metastasis [40,41,42,43]. Although this axis has been implicated in colorectal [41], breast [42], and gastric cancers [43], to our knowledge this is the first suggestion of such a relationship in HPVOPC. Therefore, we decided to start exploring this possibility by confirming axis expression in HPVOPC.

To confirm which types of immune cells express these chemokine ligands, we referenced a public database of single-cell RNA-seq (scRNA-seq) of HPVOPC data [33]. This database was derived from paired tissue and blood samples of hematological origin (CD45+ cells) from primary tumors of 18 HPV(−) and 8 HPV(+) HNSCC patients, and 5 tonsil tissue samples as a control. Based on t-distributed stochastic neighbor embedding (t-SNE) plots of these scRNA-seq using the Single-Cell Expression Atlas (https://www.ebi.ac.uk/gxa/sc/home (accessed on 1 November 2018)), we subdivided these cells into 18 clusters (Figure 4A). These cells were also divided into three groups based on the origin (Blood, Neoplasm, and Tonsil). The 18 clusters were annotated with comprehensive immune cell markers: T cell; CD3D, CD4Tcell; CD4, CD8Tcell; CD8B, B cell; CD79B, regulatory T cell; FOXP3, NK cell; CD56 (NCAM1), macrophage; CD68, monocyte cell; CD14. First, we checked the distribution of CXCL9, CXCL10, and CXCL11 expression in each cluster using a feature plot (Figure 4B and Appendix A). CXCL9, CXCL10, and CXCL11 were expressed most prominently by cluster 11, which was annotated as CD14(+) monocytes derived from neoplasms (Figure 4A–C). There were 4 clusters annotated as CD14(+) monocyte cells, but among these, cluster 11 expressed the highest levels of CXCL9, CXCL10, and CXCL11 (Figure 4D). Finally, CXCL9, CXCL10, and CXCL11 expression within cluster 11 was analyzed for macrophage (CD68), M2 macrophage (CD163), and dendritic cells (CD11C) (Figure 4E). This demonstrated significantly higher expression of CXCL9 and CXCL10 for the anti-tumor immune-suppressive M2 tumor-related macrophage family than for the M1 macrophage (*p* = 6 × 10^−27^, 8 × 10^−6^, respectively) (Figure 4F). Taken together, these in silico and in vitro analyses suggest that the CXCR3 chemokine axis may play an important role in tumor progression related to inhibition of anti-tumor immunity, but whether this role could be exploited to impede tumor growth was previously untested.

Next, immunohistochemistry was performed on primary HPVOPC tumors and benign oropharynx tissue (normal tonsils) to evaluate the expression of CXCR3 and its ligand CXCL9. HPVOPC tumors showed higher expression when compared to benign oropharynx tissue for both CXCR3 and CXCL9, and they demonstrated higher expression in the tumor itself than in the surrounding stromal cells (Figure 5A,B). To confirm the microenvironment expression status in stromal cells more precisely, we performed mIHC for a macrophage panel (CD68, CD163, and CD16). As expected, there was a higher macrophage presence for HPVOPC tumors compared to benign tissue (Figure 5C,D), and this also validated that CXCR3 and CXCL9 were highly expressed in tumor cells themselves, as well as macrophages (Figure 5A).

Finally, to determine if our hypothesis that the *CXCR3* chemokine axis is a targetable driver of tumor growth, we performed in vivo experiments with both immunodeficient and immunocompetent murine models, in which appropriate SCC tumor cell lines were implanted in mice and the tumor volume and weight were compared between the antagonist of CXCR3 (AMG487)-treated and vehicle-treated groups (Figure 6A). For the immunodeficient model, athymic nude mice and HPV16-positive squamous cell carcinoma (UM-SCC-47) tumors were utilized. The tumor volume and weight in immunodeficient animals treated with AMG487 were statistically significantly lower than in control animals (*p* = 0.001 and 0.008, Student’s *t*-test, respectively; Figure 6B–E), supporting the conclusion that the antagonist of CXCR3 inhibits the tumor growth in an immunodeficient model. Next, for the immunocompetent model, C57BL/6 mice and mEER-derived squamous cell carcinoma tumors were utilized (Figure 6F). The tumor volume and weight in immunocompetent animals treated with AMG487 did not demonstrate a significant difference compared to the control animals (Figure 6G–J), suggesting that the CXCR3 antagonist does not inhibit tumor growth in an immunocompetent model, and may have effects on the immune tumor microenvironment that offset the tumor proliferative effects of CXCR3 ligands.

## 4. Discussion

In this investigation, we examined the networks of genes associated with four specific mechanisms of alteration, including ASE, methylation, SNV, and CNV, and their overlap with the network of differentially expressed genes in HPVOPC. We found significant overlap with ASE, methylation, and SNV, but not CNV alterations, and these findings were confirmed with a network analysis using a validation cohort of tumors from TCGA. HPVOPC is known to have significantly fewer mutations per tumor than HPV-negative tumors [44] and a strikingly lower burden of mutations within targetable oncogenic pathways [4], which underscores how incompletely mutational analysis alone describes the molecular drivers of HPV-related disease. Epigenetic modifiers of expression associated with DNA methylation and post-transcriptional changes, such as RNA splicing alterations, are important mechanisms that allow for changes in gene expression, and this work supports the idea that these modifications, in addition to genetic mutations, may have a central role in driving the tumor phenotype. Previous work has also postulated the importance of alternate splicing as a functional mechanism of oncogenesis in HPVOPC [9,45], and other studies have also demonstrated unique differential methylation signatures for HPVOPC [11,46,47,48,49].

For each class of alteration, clusters of genes associated with immune system functions consistently demonstrated the highest amount of overlap with the network of DEG. Within the immune cluster of DEG, *PIK3CA* arose at the center of this subnetwork, consistent with prior studies that demonstrate *PIK3CA* to be a central disrupted pathway in HPVOPC. *PIK3CA* has been shown to be mutated in about 56% of HPVOPC tumors, often leading to overexpression [5,6]. Additionally, the *PI3K* pathway is activated in immune cells and implicated in adaptive resistance to immune checkpoint inhibitors such as anti-PD-1 [50,51]. Since our analysis was performed on data obtained from whole tumor samples, including the immune component rather than micro-dissected tumor specimen, our *PIK3CA* signal may be from the tumor cells, immune cells, or both. This analysis looks at both the tumor-specific and immune cell expression simultaneously, but we looked at each class of tumor alterations (i.e., SNV, methylation, and splice alterations) separately to deconvolute the signals. Since the clusters of genes associated with immune function consistently demonstrated the most overlap with DEG networks within every class of specific alteration, this suggests that there may be cooperative interactions between each class of tumor alterations in disrupting the ultimate network that they affect; in this case, immune regulation. These data are consistent with HPVOPC being an immune-therapy-responsive tumor, confirmed by recent data showing responsiveness to immune therapies and a moderate-to-high tumor mutational burden [52,53,54,55,56].

Additionally informative is the unexpected relative lack of an association between networks of SNV and DEGs for clusters of genes beyond solely the immune system and *PIK3CA*, suggesting that SNV drives gene expression mainly through the immune/*PIK3CA* cluster. This is entirely congruent with the high rate of *PIK3CA* network mutations, including *PIK3CA* as the most highly mutated gene in HPVOPC [5,6]. In our analysis, the finding of *PI3CKA* centrally located within the “immune system” cluster of the network space differentially expressed genes was a reassuring finding considering the *PI3K* pathway regulates multiple cellular processes underlying the immune responses against malignant cells and has previously been implicated in HPVOPC. Although *PIK3CA* is consistently found within the subnetworks for immune clusters, it is important to remember that because *PIK3CA* is such a dominantly disrupted pathway in HPVOPC, its presence at the center of immune clusters may simply represent a tumor-intrinsic *PI3K-AKT* signature, rather than a strictly immune signature.

The association of *PIK3CA* and immune network alteration with SNV contrasts with other alteration classes, which drive distinct alternate gene networks. For example, methylation and splicing both had significant overlap for clusters involved with G-alpha (i) signaling events, keratinization, and others. This could suggest that different classes of alterations are responsible for the alteration of different gene networks to drive DEGs and, ultimately, the malignant tumor phenotype.

Classically, whole-exome association studies in HPVOPC have concluded that SNVs in specific genes, such as within the *PIK3CA* pathway, largely account for the malignant tumor phenotype [6]. These data demonstrate that network methylation alterations are more closely associated with changes to the overall network gene expression in HPVOPC than other tumor alteration mechanisms, which suggests epigenetic changes are more responsible for driving differential gene expression dysregulation in this cancer than previously thought. The relatively high association of ASE with global differential gene expression further confirms that epigenetic and transcriptional alterations are as important as genomic mutations in driving tumor-specific differential gene expression, if not more so.

Network analyses with our discovery cohort demonstrated significant amount of overlap in clusters representing genes involved with the GPCR signaling pathways G-alpha (q) and G-alpha (i) between the DEG network and individual alteration-type networks (splicing, methylation, and CNV for G-alpha (i), and methylation and CNV for G-alpha (q)). GPCRs are well-established participants in pathways for many key physiologic functions and are implicated in cancer invasion, metastasis, angiogenesis, and immune evasion [57,58]. G-alpha (q) has been shown to be important in cytoskeletal-mediated cellular responses in epithelial cells with the emergence of a malignant phenotype arising from both under- and over-expression of G-alpha-11/q signaling [59]. Recent efforts by Wu et al. have described a network of Onco–GPCRome interactions promoting tumor growth, dissemination, and immune evasion, stressing their potential roles as targets in the new era of precision medicine and oncogenic immunotherapy [60,61].

Interestingly, the trend of overlap between network alteration classes changes when the significance threshold is elevated from an alpha of 0.05 to less than 0.001. Network changes within alternative splicing have a higher fraction of extremely significant genes with absolute log-fold expression changes greater than 2 overrepresented in the differential gene expression network than other tumor alteration classes. It is important to note that this list of alternatively spliced genes has been extensively curated and validated, as described in previous publications, and may, therefore, be highly enriched for biologically significant alterations [9]. The tight association of ASE with differentially expressed genes compared to other alteration classes suggests that this mechanism may be more responsible for the overall malignant phenotype in HPVOPC than previously thought. This is congruent with recent data suggesting that *APOBEC* mediated transcriptional alterations in a key carcinogenic mechanism of multiple solid tumors [62,63], and indeed HPV- and non-HPV-related HNSCC have a key *APOBEC* signature [64]. Another important finding is that genes significant in the CNV network space are less associated with the network space’s overall gene expression than other alteration mechanisms. CNV is often considered an important driver of differential expression in tumors, but this analysis suggests its role may ultimately be less central at the expression level in HPVOPC.

Our analysis of TCGA dataset largely validated our findings from the discovery cohort. As shown in Figure 4, however, there were differences in the relative overrepresentation of DEG in network subgraphs, with greater overrepresentation found in SNV for TCGA, though methylation and splicing associations were still significant. This discrepancy may reflect methodological differences in how the primary alteration data were obtained. For example, TCGA methylation analysis was performed with HumanMethylation450 arrays, which have much lower resolution than that for the discovery cohort, which used MBDseq, allowing 100 bp segment resolution. Additionally, the curation of specific alternative splicing events in the discovery cohort utilized a larger sample of normal mucosa (n = 25), but that from TCGA validation cohort was smaller (n = 16). Despite this modestly sized cohort, it is remarkable that these alternative splicing events were still significant, and very highly associated with global DEG.

The finding of a GPCR, specifically G-alpha (i) and G-alpha (q), methylation signal after network analysis has important implications for the tumor microenvironment in HPV-OPSCC. Among GPCRs differentially expressed in both the discovery and validation cohorts that interacted with HPV16 oncoproteins E6 and E7, there was a strong signal for chemokine signaling (Figure 3B,C), specifically *CXCL9*, *CXCL10*, and *CXCL11*. The *CXCL9*, *10*, and *11/CXCR3* chemokine axis has been increasingly recognized as an important regulator of the tumor microenvironment [36,37]. Classically, this chemokine axis is thought to regulate immune cell migration, differentiation, and activation in a paracrine fashion and play an anti-tumor role in tumor suppression and immune surveillance [38,39]. More recently, however, the same *CXCR3* chemokine axis has also been shown to have paradoxically a pro-tumor effect, in which, in an autocrine signaling fashion, it increases tumor cell proliferation, angiogenesis, and metastasis [40,41,42,43]. We tested the hypothesis that the *CXCR3* chemokine axis has a pro-tumor effect in HPVOPC and is a targetable driver of tumor growth in an athymic mouse model. Animals exposed to the CXCR3 antagonist indeed had slower tumor growth than control animals with an intact *CXCR3* chemokine axis, supporting the conclusion that antagonism of CXCR3 inhibits tumor growth. Our mutational network analysis (SNV) did not find such GPCRs to play as significant role as they did in splicing and methylation aberrations, which highlights the utility of our novel multi-platform “omics” approach.

Our analysis of scRNA-seq data revealed that HPV-associated OPC tumor microenvironment expression of this chemokine axis was highest for monocyte cells, and higher for M2 macrophages compared to M1 macrophage populations. M2 macrophage populations have been shown to establish a pro-tumor environment [65,66], making this axis worthy of further investigation as a potential therapeutic target. Related work has shown *CXCR3* and its ligands to be among a host of genes whose expression is upregulated in HPVOPC via the increased activity of *PI3K gamma*, which functions as a molecular switch that promotes immunosuppression during tumor growth [67]. Although the *CXCR3* axis has been implicated in osteosarcoma [68], colorectal [41], breast [42], cervical [69], and gastric cancers [43], to our knowledge this is the first suggestion of such a relationship in HPVOPC. Cambien and colleagues demonstrated, in both immunosuppressed and immunocompetent murine models, that *CXCR3* is induced by CXCL9/10/11 and mediates colorectal tumor cell metastasis as well as tumor cell survival and progression after implantation, and that systemic CXCR3 antagonism significantly reduces metastasis to the lung [41]. Zhu and colleagues showed, in a murine breast cancer model, that CXCR3 acts both on the tumor and host compartments by promoting metastasis and impairing the host’s anti-tumor immunity, and that both could be improved through chemical and genetic approaches to *CXCR3* inhibition [42]. Although the CXCR3 chemokine axis is not a specific feature of HPV-related tumors, a recent publication investigating the immune signature differences between HPV-related and HPV-negative OPC did demonstrate a relative upregulation of CXCR3 in HPVOPC [70]. Chen and colleagues have demonstrated, in other HPV-driven cancers, specifically HPV-positive cervical cancer, that CXCL10 is upregulated and its receptor CXCR3 is overexpressed [69]. Prior work found increased CXCR3 expression in about 60% of samples on a tissue microarray of HNSCC, though not specifically oropharynx subsite or HPV-related tumors [71]. Herein, we demonstrated that antagonism of CXCR3 in HPVOPC significantly slowed tumor growth in an immunodeficient model in which the pro-tumor autocrine effect of the CXCR3 axis dominates. However, in an immunocompetent model in which the classical immune-mediated, anti-tumor paracrine effect of the CXCR3 axis was restored, antagonism of CXCR3 had no significant difference compared to the control animals, in line with previous studies implying a dysfunctional *CXCR3* axis implicated in both anti-tumor immune modulation of the tumor microenvironment, as well as the opposing action of pro-tumor growth simulation. Taken together, these results suggest that CXCR3 inhibition in oropharynx squamous cell carcinoma should be used with caution, as these agents may stimulate tumor growth considering the competing effects of this cytokine axis. Further work is ongoing to corroborate our hypothesis-generating findings that the *CXCR3* autocrine axis may drive tumor progression in HPVOPC and to elucidate the mechanisms by which it may interact with HPV proteins. It is worthwhile to acknowledge the present effort to identify opportunities for treatment de-escalation in majority of carefully selected patients with HPVOPC. However, additional treatment strategies might benefit patients whose tumors display a more aggressive phenotype either in the treatment-naive or recurrent disease setting.

## 5. Conclusions

Methylation, ASE, and SNV alterations were highly associated with network gene expression changes in HPVOPC, suggesting that ASE and methylation alterations have an important role in driving the oncogenic phenotype. Network analysis identified GPCR networks, specifically the CXCR3 chemokine axis, as modulators of tumor–immune interactions that may have proliferative effects on primary tumors, as well as a role for immunosurveillance, and may present opportunities for targeted therapy.

## Figures and Tables

**Figure 1 cancers-15-04379-f001:**
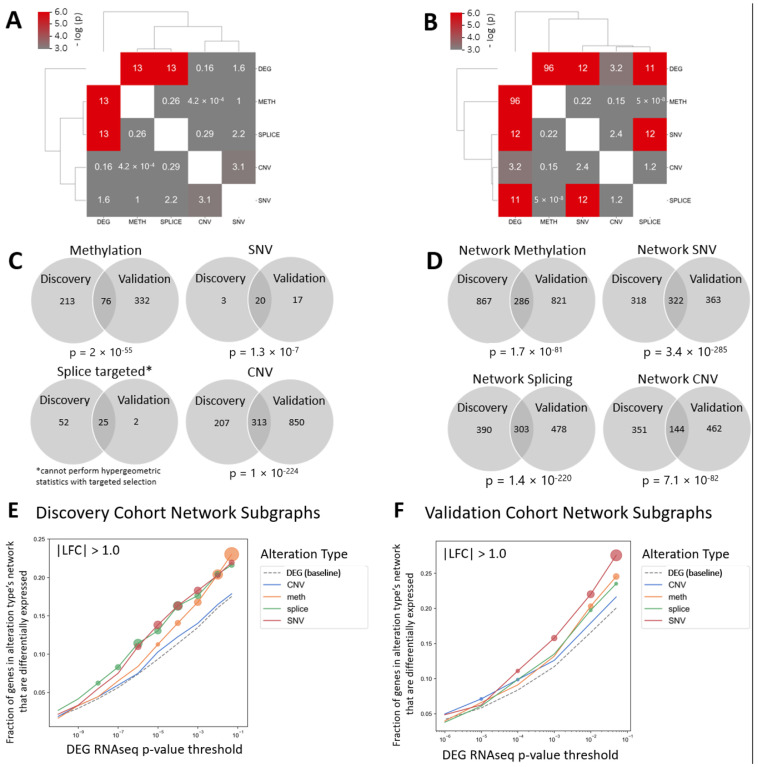
Network propagation reveals significant gene overlap among all tumor alteration types for differentially expressed genes, with significant enrichment among splicing, methylation, and SNV alteration networks. (**A**) Heatmap showing the overlap of altered seed genes for each alteration type in the discovery cohort before network propagation. Values > 3.0 meet significance with hypergeometric testing. (**B**) Heatmap showing results of the same overlap calculation following network propagation. (**C**,**D**) Venn diagrams of gene set overlap by alteration type between the discovery and validation cohorts both before network propagation (**C**) and after network propagation and exclusion of seed genes (**D**). To visualize the enrichment for differentially expressed genes within each tumor alteration type after network propagation analysis, the fraction of genes for each alteration type’s network subgraph that are differentially expressed (LFC magnitude of 1 or greater) was plotted against varying *p*-value significance thresholds for differential expression (RNAseq) for: (**E**) the discovery cohort and (**F**) the validation cohort. The bubble size is proportional to the increasing statistical significance of enrichment for DEGs among each alteration type’s network subgraph. DEGs = differentially expressed genes, METH = methylation, SPLICE = alternative splice events, CNV = copy number variation, SNV = single-nucleotide variation, LFC = log-fold change in gene expression.

**Figure 2 cancers-15-04379-f002:**
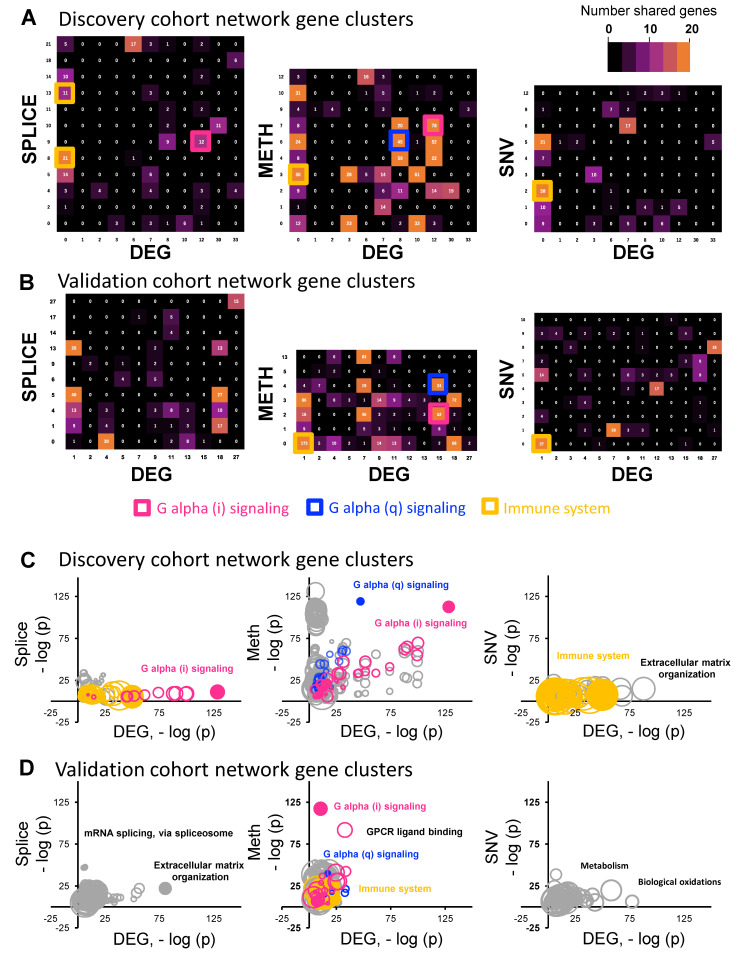
Network enrichment analysis shows clusters of genes associated with GPCR signaling and the immune system. Heatmaps of annotated gene clusters following enrichment analysis visualizing significant overlap in the DEG network and the other tumor alteration networks for (**A**) the discovery cohort and (**B**) the validation cohort. Number of shared pathways is shown if ≥2. Pink, blue, and gold squares highlight overlapping gene clusters of G-alpha (i) signaling, G-alpha (q) signaling, and the immune system, respectively. (**C**,**D**) Bubble plots graphing the *p*-value of genes within the cluster of each tumor alteration type against that for DEG clusters for the discovery cohort (**C**) and the validation cohort (**D**). Pathways that are plotted toward the top right of the axes are differentially altered, with strong statistical association. The bubble size is proportional to the number of overlapping genes in each shared cluster, and the bubble color corresponds to the same gene clusters illustrated in part (**A**,**B**) above. DEG = differentially expressed genes, METH = methylation, SPLICE = alternative splice events, SNV = single-nucleotide variation.

**Figure 3 cancers-15-04379-f003:**
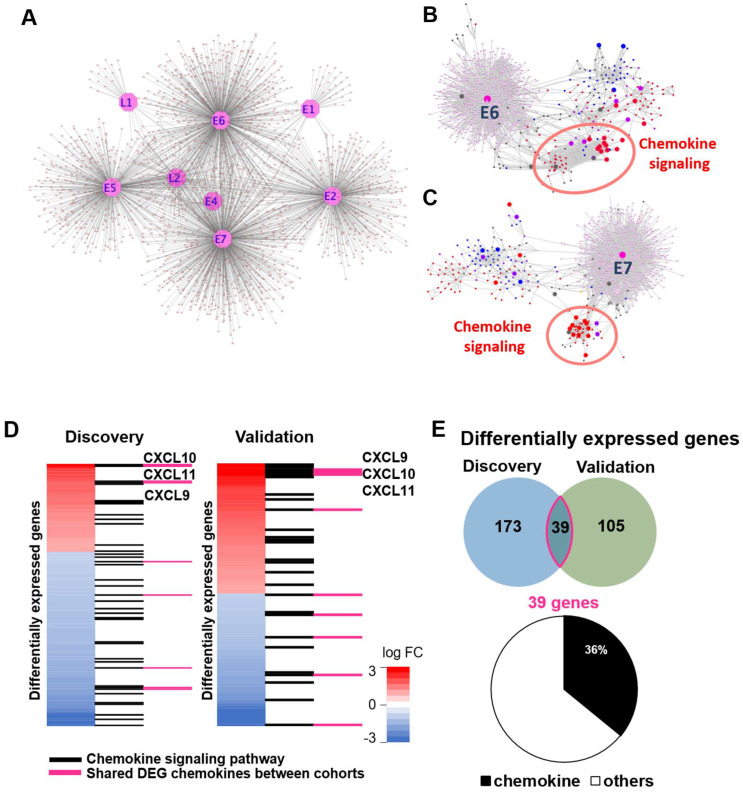
Chemokines interact with HPV16 viral proteins and are overexpressed. Cytoscape visualizations of publicly available data of interactions between HPV16 viral proteins and human proteins in general (**A**) and of interactions between HPV16 oncoproteins E6 (**B**) and E7 (**C**) and known human GPCR signaling proteins (red dots for proteins annotated in reactome as G-alpha (i), blue dots for G-alpha (q), and purple dots for proteins annotated as both). Red circles emphasize remarkable interactions between viral oncoproteins E6/E7 and chemokine signaling proteins. (**D**) Examination of genes included in GPCR signaling pathways confirmed a higher tumor expression pattern of CXCR3-axis chemokine signaling in both the discovery and validation cohorts, including *CXCL9*, *CXCL10*, and *CXCL11*. (**E**) Venn diagram showing overlap of differentially expressed genes between the discovery and validation cohorts (**above**) and pie chart showing the proportion of overlapping genes that play a role in chemokine signaling (**below**).

**Figure 4 cancers-15-04379-f004:**
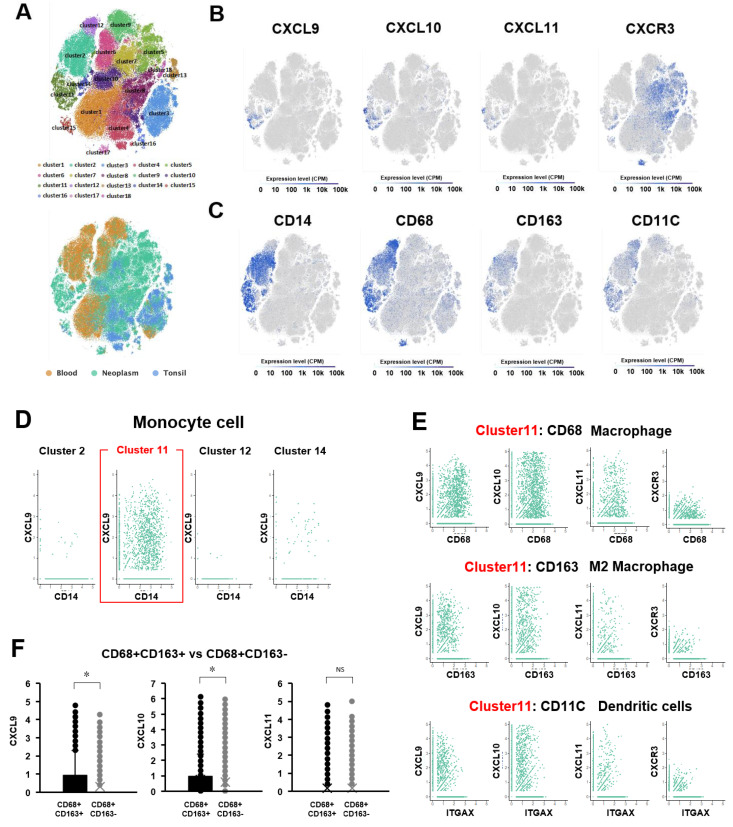
*CXCL9*, *10*, and *11/CXCR3* axis expression among immune cell populations in HPVOPC shows the highest *CXCL9* and *10* expression by M2 macrophages. (**A**) t-SNE plots of publicly available scRNA-seq HPVOPC data subdivided into 3 groups based on origin (blood, neoplasm, tonsil control; below) and into 18 clusters using the publicly available Single-Cell Expression Atlas. (**B,C**) The same t-SNE plot showing the distribution in each cluster of *CXCL9/10/11* and *CXCR3* expression (**B**) or immune cell markers for monocytes (CD14), macrophage (CD68), M2 macrophage (CD163), and dendritic cells (CD11c) (**C**). (**D**) Plots of CD14(+) monocytes from several clusters showing *CXCL9* expression. A red rectangle highlights the higher expression found in cluster 11. (**E**) Plots showing the expression of *CXCL9/10/11* and *CXCR3* in cells from cluster 11, including macrophage (CD68, top), M2 macrophage (CD163, middle), and dendritic cells (CD11c, bottom). (**F**) Graphs comparing the expression of *CXCL9/10/11* between M2 and M1 macrophage populations shows significantly higher expression of *CXCL9* and *10* by M2 macrophages. “*” denotes that the difference between groups reach statistical significance, whereas “NS” denotes a failure to reach statistical significance.

**Figure 5 cancers-15-04379-f005:**
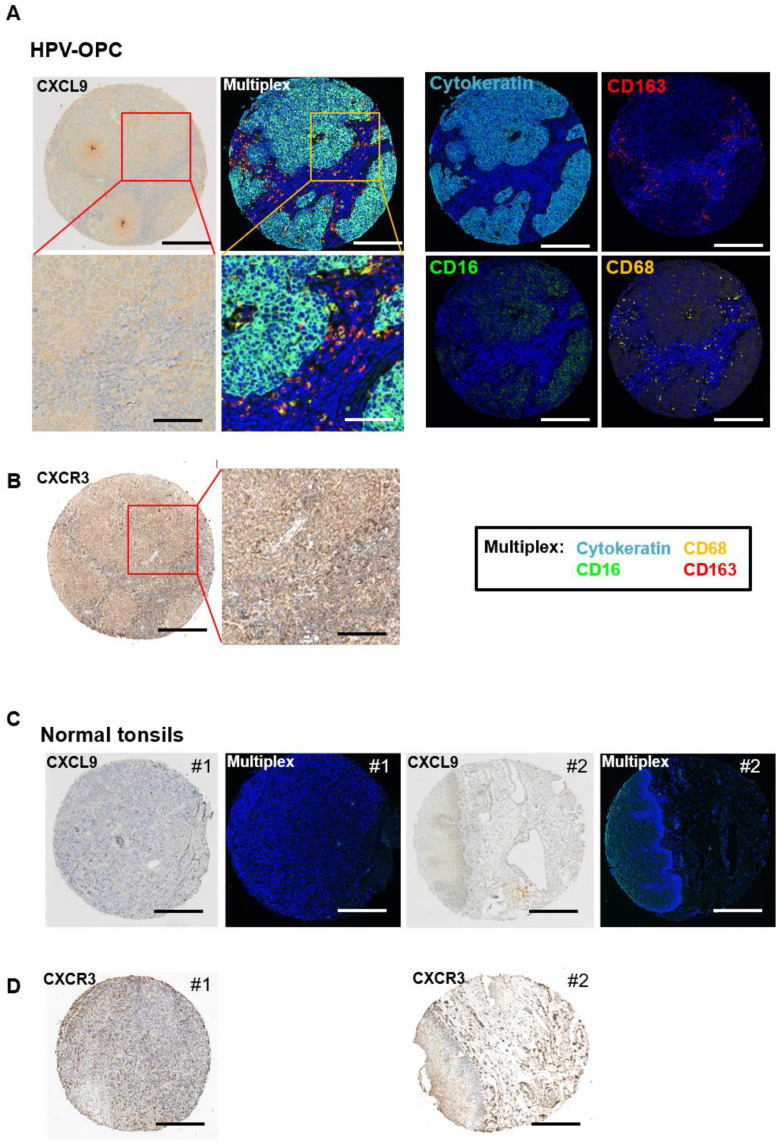
HPVOPC primary tumors express CXCR3 and CXCL9 higher than benign oropharynx tissue. Immunohistochemistry comparing the expression of CXCR3 and CXCL9 between primary HPVOPC tumors (**A**,**B**) and benign oropharynx tissue #1 and #2 (normal tonsils, **C**,**D**). The scale is 200 μm, and the expanded scale (in red and yellow rectangles) is 80 μm, respectively. Multiplex immunohistochemistry with cytokeratin and various macrophage markers (CD68, CD163, and CD16) also demonstrates a higher macrophage presence in HPVOPC tumors compared to normal tissue, as expected. Individual picture of each antibody in multiplex immunohistochemistry of HPVOPC is shown in (**A**, right).

**Figure 6 cancers-15-04379-f006:**
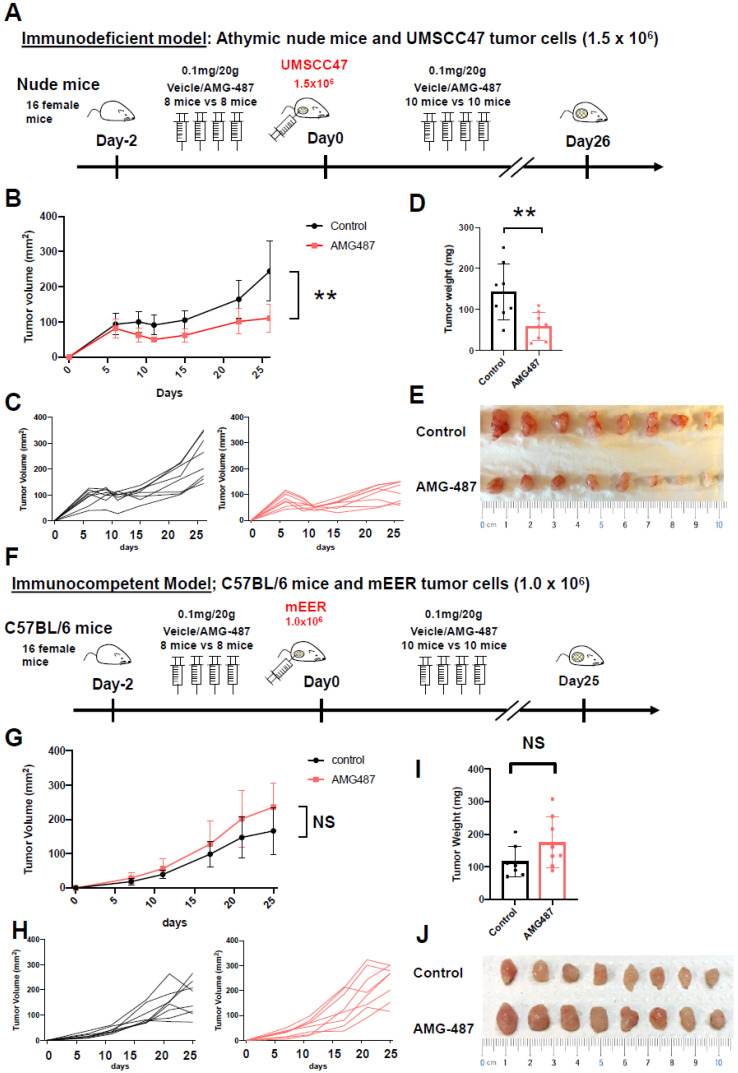
In vivo CXCR3 antagonism inhibits tumor growth in immunodeficient but not immunocompetent mouse models. (**A**) Experiment schematic for immunodeficient model: briefly, 16 female nude mice were treated with the CXCR3 antagonist AMG-487 or vehicle two days before implantation with UMSCC47 tumor cells, and then treated similarly daily until euthanized at day 26 for tumor measurement. (**B**–**D**) Immunodeficient model using athymic nude mice and UMSCC47 tumor cells (1.5 × 10^6^). Tumor volume over time, showing average among groups (**B**) and curves for individual animals (**C**). Tumor weight compared between groups on day 26 (**D**). Photograph of excised tumors on day 26 for animals in both groups (top: vehicle, bottom: AMG-487 group) (**E**). By both volume and weight, tumors were significantly smaller in animals administered the CXCR3 antagonist. (**F**) Experiment schematic for the immunocompetent model: similar to the immunodeficient model (**A**) except C57BL/6 mice and mEER tumor cells were used and treated until euthanized at day 25. (**G**–**J**) Immunocompetent model using C57BL/6 mice and mEER tumor cells (1.0 × 10^6^). Tumor volume over time, showing average among groups (**G**) and curves for individual animals (**H**). Tumor weight compared between groups on day 26 (**I**). Photograph of excised tumors on day 26 for animals in both groups (top: vehicle, bottom: AMG-487 group) (**J**). By both volume and weight, there was no significant difference between the vehicle and the CXCR3 antagonist. “**” denotes that the difference between groups reach statistical significance, whereas “NS” denotes a failure to reach statistical significance.

**Table 1 cancers-15-04379-t001:** Multiplex immunohistochemistry panel parameters.

Antigen Retrieval	Antibody	Clone	Dilution	Vendor	Incubation (min)	2^0^ HRP	TSA-Opal	Dilution
AR1 (20 min)	CD16	2H7	1:50	LSbio, Lynnwood, WA, USA	30 min	Opal Polymer	570	1:150
AR1 (20 min)	CD163	MRQ-26	Pre-dilute	Ventana, Oro Valley, AZ, USA	45 min	Opal Polymer	690	1:150
AR1 (20 min)	CD68	PGM-1	1:400	Dako, Glostrup, Denmark	60 min	Opal Polymer	620	1:150
AR1 (20 min)	Cytokeratin	AE1/AE3		Dako	30 min	Opal Polymer	520	1:150

## Data Availability

The data that support the findings of this study are available from the corresponding author upon reasonable request. Sequencing data have been previously deposited and are available at the following database locations (accessed on 1 November 2018): https://www.ncbi.nlm.nih.gov/geo/query/acc.cgi?acc=GSE112021, https://www.ncbi.nlm.nih.gov/geo/query/acc.cgi?acc=GSE112023, https://www.ncbi.nlm.nih.gov/geo/query/acc.cgi?acc=GSE112026, https://www.ncbi.nlm.nih.gov/geo/query/acc.cgi?acc=GSE112027.

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
