# Peer review of "A Network Landscape of HPVOPC Reveals Methylation Alterations as Significant Drivers of Gene Expression via an Immune-Mediated GPCR Signal"

_cancers, 2023, doi:10.3390/cancers15174379_

Round 1
Reviewer 1 Report
This work by Qualliotine et al. uses sophisticated data/network analysis to look at multiple classes of tumor alterations in a cohort of 46 primary HPV-associated OPC tumors and 25 controls (cancer unaffected). They validated their analysis findings with TGCA data, identifying overlap between DEG networks and alteration classes including DNA methylation and copy number variance. Their protein interaction analysis identified enrichment of immune- GPCR signaling hubs, including CXCR3 and cytokine ligands CXCL9, CXCL10, and CXCL11. The receptor CXCR3 was found to be expressed in HPV+ OPC, and scRNA-seq demonstrated CXCR3 ligands to be highly expressed in M2 macrophage.
An immunodeficient murine model using UMSCC47 cells demonstrated decreased tumor growth with antagonism of the CXCR3 receptor by small molecule antagonist AMG-487, suggesting that the CXCR3 axis can drive tumor proliferation, albeit in the context of athymic nude mice. In contrast, AMG-487 treatment in an immunocompetent C57BL/6 model with mEER tumors (murine keratinocytes transformed with HPV16 E6/E7 oncogenes) showed no significant differences in tumor growth, and tumors were actually larger with treatment (although this wasn't statistically significant).
Overall, the data and analysis identify GPCR networks and the CXCR3 chemokine axis as potential modulators of tumor-immune interactions that may drive proliferation of primary tumors. The data with immunocompetent C57BL/6-mEER model was essentially ignored in the abstract and they need to back off the strong conclusions based on the one in vivo study performed in immunodeficient athymic nude mice.
Are the differences between these two models due to the different cells used (UMSCC47 vs mEER)? Do these two cell lines express CXCR3 to comparable levels? Do species differences between human and mouse CXCR3 account for any differences? How would the mEER tumors respond to the antagonist in athymic mice?
Reviewer 2 Report
The manuscript from Qualliotine et al addresses the biology behind HPV related OPC through an integrative approach including methylation, ASE, and CNV. CXCR3 has been already investigated in HNC field; this could affect the novelty of the work. The experimental design and methodology are sound. However, the manuscript is quite complex to follow. A graphical abstract is recommended to help the readers through the analyses.
Here below the main points that deserve to be explained
The clinical tables showing the main parameters of both training and validation sets (i.e. demographics, TNM, smoking, alcohol, treatment, etc) are lacking. Are the cohort comparable? Does the TCGA (n=94) include only OPC?
The starting point of this work is a comparison between normal and tumor tissues; how does the class imbalance (training 46 vs 25; validation 94 vs 16) affect the results? It’s unclear how the bioinformatics analyses address the issue.
Are CXCR3 and the other genes of the chemokine axis a specific feature of HPV-related tumors? Or present also in HPV-negative OPC?
The variability of CXCR3 expression in HPV tumors is not shown. Theoretically speaking, if anti-CXCR3 will be moved in a real clinical setting, are all patients eligible for the treatment? Or we should expect to treat only a subset of patient (i.e. CXCR3 expressing)? Just in case, could you give a hint on the % of CXCR3 expressing tumors?
Treatment de-escalation is the main clinical need in HPV positive disease. It’s really unclear how Qualliotine et al findings could fit in the current treatment schemes of HPV-related OPC.
Why do you choose female mice?
Could the different behavior of CXCR3 antagonist in athymic murine model compared to immunocompetent animals be associate to the known HPV molecular subtpyes where different immnune-landscapes were found?
Sequencing data are not made freely available to the scientific community; however, following the journal research data policies (https://www.mdpi.com/journal/cancers/instructions#suppmaterials), new sequence information must be deposited to the appropriate database prior to submission of the manuscript. Accession numbers provided by the database should be included in the submitted manuscript.
This reviewer had not access to the supplementary material.
Reviewer 3 Report
The article entitled “A Network Landscape of HPVOPC Reveals Methylation Alterations as Significant Drivers of Gene Expression via an Immune-Mediated GPCR Signal” by Qualliotine et al., demonstrated the role of methylation, alternative splicing events and single nucleotide variation in driving oncogenic phenotypes in HPV associated oropharynx carcinoma. These alternations were demonstrated to be associated with changes in the gene expression during HPV associated OPC. Further, the authors identified that gene clusters of G protein coupled receptor (GPCR) pathways, more specifically CXCR3 chemokine axis, as modulator of tumor-immune interactions having an effect on primary tumors.
The research was carried out with clear objectives and well planned methodology. The results and conclusions from the research is well described.
The MS is meeting the standards of the journal “CANCERS” not only in terms of the quality of research work but also in terms of writing and presentation.
The article can be accepted for publication in the “CANCERS” in present form.
